# Phylogenetic and Taxonomic Analyses of Three New Wood-Inhabiting Fungi of *Xylodon* (Basidiomycota) in a Forest Ecological System

**DOI:** 10.3390/jof8040405

**Published:** 2022-04-15

**Authors:** Kai-Yue Luo, Zhuo-Yue Chen, Chang-Lin Zhao

**Affiliations:** 1Key Laboratory for Forest Resources Conservation and Utilization in the Southwest Mountains of China, Ministry of Education, Southwest Forestry University, Kunming 650224, China; fungikaiyuel@163.com; 2Yunnan Key Laboratory of Plateau Wetland Conservation, Restoration and Ecological Services, Southwest Forestry University, Kunming 650224, China; 3College of Biodiversity Conservation, Southwest Forestry University, Kunming 650224, China; fungizhuoyuechen@163.com; 4Yunnan Key Laboratory for Fungal Diversity and Green Development, Kunming Institute of Botany, Chinese Academy of Sciences, Kunming 650201, China; 5School of Life Sciences, Tsinghua University, Beijing 100084, China

**Keywords:** biodiversity, ecology, environment, molecular phylogeny, white rot fungi, Yunnan–Guizhou Plateau

## Abstract

Wood-inhabiting fungi are a cosmopolitan group and show a rich diversity, growing in the vegetation of boreal, temperate, subtropical, and tropical regions. *Xylodon grandineus*, *X. punctus*, and *X. wenshanensis* spp. nov. were found in the Yunnan–Guizhou Plateau, China, suggested here to be new fungal species in light of their morphology and phylogeny. *Xylodon grandineus* is characterized by a grandinioid hymenophore and ellipsoid basidiospores; *X. punctus* has a membranous hymenophore, a smooth hymenial surface with a speckled distribution, and absent cystidia; *X. wenshanensis* has a grandinioid hymenophore with a cream to slightly buff hymenial surface and cystidia of two types. Sequences of the ITS and nLSU rRNA markers of the studied samples were generated, and phylogenetic analyses were performed using the maximum likelihood, maximum parsimony, and Bayesian inference methods. After a series of phylogenetic studies, the ITS+nLSU analysis of the order Hymenochaetales indicated that, at the generic level, six genera (i.e., *Fasciodontia*, *Hastodontia*, *Hyphodontia*, *Lyomyces*, *Kneiffiella*, and *Xylodon*) should be accepted to accommodate the members of *Hyphodontia* sensu lato. According to a further analysis of the ITS dataset, *X. grandineus* was retrieved as a sister to *X. nesporii*; *X. punctus* formed a monophyletic lineage and then grouped with *X. filicinus*, *X. hastifer*, *X. hyphodontinus*, and *X. tropicus*; and *X. wenshanensis* was a sister to *X. xinpingensis*.

## 1. Introduction

In forest ecosystems, fungi play essential ecological roles, driving carbon cycling in forest soils, mediating mineral nutrition of plants, and alleviating carbon limitations [1]. Wood-inhabiting fungi are a cosmopolitan group and have a rich diversity related to the high diversity of plants growing in boreal, temperate, subtropical, and tropical regions [2,3,4,5,6,7,8,9]. The order Hymenochaetales Oberw. comprises many representative wood-inhabiting fungal taxa, including hydnoid, corticioid, and polyporoid fungi possessing basidiomes with diverse hymenophoral and cystidial morphology [10,11,12,13,14]. Members of the family Schizoporaceae Jülich are widely found in different countries and areas. In addition, they cause white rot [15].

To accomplish the genome evolution and reconstruction of the phylogenetic relationships of fungi, an increasing number of taxa have been used for the fungal tree of life by using genome-scale data in the molecular systematics by mycologists [16], and both the species diversity and the classification of fungi are still in great, flux mainly in the more basal branches of the tree topology. The true diversity will come to light from genomic analyses and more region surveys worldwide based on some unique fungal groups.

The wood-inhabiting fungal genus *Xylodon* (Pers.) Gray (Schizoporaceae, Hymenochaetales) is typified by *X. quercinus* (Pers.) Gray [4]. This genus is characterized by the resupinate or effuse basidiomata with a smooth, tuberculate, grandinioid, odontioid, coralloid, irpicoid, or poroid hymenophore; a monomitic or dimitic hyphal system with clamped generative hyphae; the presence of different types of cystidia; utriform or suburniform basidia; and cylindrical to ellipsoid to globose basidiospores, in addition to causing white rot [4,17]. Based on the MycoBank database (http://www.mycobank.org, accessed on 20 March 2022) and the Index Fungorum (http://www.indexfungorum.org, accessed on 20 March 2022), the genus *Xylodon* has registered 218 specific and infraspecific names, but the actual number of the species has reached 92 [4,5,12,14,18,19,20,21,22,23,24,25,26,27,28,29,30,31,32,33,34,35,36,37,38,39,40,41,42,43].

These pioneering studies of the genus *Xylodon* were just the prelude to the molecular systematics period [16]. *Hyphodontia* s.l. was shown to be a polyphyletic genus, in which *Xylodon* and *Kneiffiella* P. Karst are the most species rich [10,12,14]. Due to a lack of rDNA sequences for many taxa, the molecular data were not enough to separate many genera clearly; therefore, a broad concept of *Hyphodontia* s.l. was employed by mycologists [10,12,14,32,34]. Yurchenko et al. described two clades: the *Xylodon-Lyomyces-Rogersella* clade and the *Xylodon-Schizopora-Palifer* clade, and they suggested to mix the species of *Xylodon*, *Schizopora* Velen., *Palifer* Stalpers and P.K. Buchanan, *Lyomyces* P. Karst., and *Rogersella* Liberta and A.J. Navas within both clades. The research comprised the representative sequences and taxa of *Hyphodontia* s.l., such as *Xylodon*, *Schizopora*, *Palifer*, *Lyomyces*, and *Rogersella*, in which the result demonstrated that it was hard to distinguish the two genera *Xylodon* and *Schizopora* on the basis of the morphological and phylogenetic information; therefore, the authors proposed that *Xylodon* and *Schizopora* should be united into the genus *Xylodon* [12]. For the phylogenetic relationship of the *Xylodon* species, it was confirmed that the two genera *Lagarobasidium* Jülich and *Xylodon* should be synonymous based on molecular data from the ITS and nLSU regions, in which the three species *X. pumilius* (Gresl. and Rajchenb.) K.H. Larss., *X. magnificus* (Gresl. and Rajchenb.) K.H. Larss., and *X. rickii* (Gresl. and Rajchenb.) K.H. Larss. were combined into *Xylodon* [36]. All of the members of the genera *Odontipsis* Hjortstam and Ryvarden and *Palifer* were placed in the genus *Xylodon* based on the molecular analyses of 28S and ITS data, in which they proposed four new species of *Xylodon* as *X. exilis* Yurchenko, Riebesehl and Langer, *X. filicinus* Yurchenko and Riebesehl, *X. follis* Riebesehl, Yurchenko and Langer, and *X. pseudolanatus* Nakasone, Yurchenko and Riebesehl [14]. Based on the multiple loci in *Hyphodontia* s.l., *Fasciodontia* Yurchenko and Riebesehl, *Hastodontia* (Parmasto) Hjortstam and Ryvarden, *Hyphodontia* J. Erikss., *Lyomyces*, *Kneiffiella*, and *Xylodon* in the order Hymenochaetales, they were divided into four clades [41]. The phylogeny of the *Xylodon* species based on the ITS and nLSU sequences proposed three new taxa from China, in which *X. gossypinus* C.L. Zhao and K.Y. Luo and *X. brevisetus* (P. Karst.) Hjortstam and Ryvarden grouped together [40]. Based on the morphological descriptions and molecular analyses, three new species: *Xylodon angustisporus* Viner and Ryvarden, *X. dissiliens* Viner and Ryvarden, and *X. laxiusculus* Viner and Ryvarden, were described and placed in *Xylodon*, which were found in Africa [42]. A phylogenetic and taxonomic study on *Xylodon* (Hymenochaetales) described three new species of this genus from southern China, inferred from 61 fungal specimens representing 55 species, which enriched the fungal diversity of this areas [43].

During investigations on wood-inhabiting fungi in the Yunnan–Guizhou Plateau of China, three additional *Xylodon* species were collected. To clarify the placement and relationships of the three species, we study carried out a phylogenetic and taxonomic study on *Xylodon*, based on the ITS and nLSU sequences.

## 2. Materials and Methods

### 2.1. Sample Collection and Herbarium Specimen Preparation

The fresh fruiting bodies of the fungi growing on fallen angiosperm branches and fallen *Pinus armandii* branches were collected from Honghe, Wenshan, and Yuxi of Yunnan Province, China. The samples were photographed in situ, and fresh macroscopic details were recorded. Photographs were recorded by a Jianeng 80D camera. All of the photos were focus stacked and merged using Helicon Focus software. Macroscopic details were recorded and transported to a field station where the fruit body was dried on an electronic food dryer at 45 °C. Once dried, the specimens were sealed in an envelope and zip-lock plastic bags and labeled [43]. The dried specimens were deposited in the herbarium of the Southwest Forestry University (SWFC), Kunming, China.

### 2.2. Morphology

The macromorphological descriptions were based on field notes and photos captured in the field and lab. The color terminology follows that of Petersen [44]. The micromorphological data were obtained from the dried specimens after observation under a light microscope with a magnification of 10 × 100 oil [27]. The following abbreviations are used: KOH = 5% potassium hydroxide water solution, CB− = acyanophilous, IKI− = both inamyloid and indextrinoid, L = mean spore length (arithmetic average for all spores), W = mean spore width (arithmetic average for all spores), Q = variation in the L/W ratios between the specimens studied, and *n* = a/b (number of spores (a) measured from given number (b) of specimens).

### 2.3. Molecular Phylogeny

The CTAB rapid plant genome extraction kit-DN14 (Aidlab Biotechnologies Co., Ltd., Beijing, China) was used to obtain genomic DNA from the dried specimens according to the manufacturer’s instructions. The nuclear ribosomal ITS region was amplified with ITS5 and ITS4 primers [45]. The nuclear nLSU region was amplified with the LR0R and LR7 primer pair (http://lutzonilab.org/nuclear-ribosomal-dna/, accessed on 22 January 2022). The PCR procedure for ITS was as follows: initial denaturation at 95 °C for 3 min, followed by 35 cycles at 94 °C for 40 s, 58 °C for 45 s and 72 °C for 1 min, and a final extension of 72 °C for 10 min. The PCR procedure for nLSU was as follows: initial denaturation at 94 °C for 1 min, followed by 35 cycles at 94 °C for 30 s, 48 °C for 1 min and 72 °C for 1.5 min, and a final extension of 72 °C for 10 min. The PCR products were purified and sequenced at Kunming Tsingke Biological Technology Limited Company (Kunming, China). All of the newly generated sequences were deposited in NCBI GenBank (https://www.ncbi.nlm.nih.gov/genbank/, accessed on 22 January 2022) (Table 1).

The sequences were aligned in MAFFT version 7 [56] using the G-INS-i strategy. The alignment was adjusted manually using AliView version 1.27 [57]. The dataset was aligned first, and then the sequences of ITS and nLSU were combined with Mesquite version 3.51. The alignment datasets were deposited in TreeBASE (submission ID 29411). ITS+nLSU sequences and ITS-only datasets were used to infer the position of the three new species in the genus *Xylodon* and related species. Sequences of *Hymenochaete cinnamomea* (Pers.) Bres. and *H. rubiginosa* (Dicks.) Lév. retrieved from GenBank were used as an outgroup in the ITS+nLSU analysis (Figure 1); sequences of *Lyomyces orientalis* Riebesehl, Yurch. and Langer, and L. *sambuca* (Pers.) P. Karst. retrieved from GenBank were used as an outgroup in the ITS analysis (Figure 2) [41].

Maximum parsimony (MP), maximum likelihood (ML), and Bayesian inference (BI) analyses were applied to the combined three datasets following a previous study [58], and the tree construction procedure was performed in PAUP* version 4.0b10 [59]. All of the characters were equally weighted, and gaps were treated as missing data. Using the heuristic search option with TBR branch swapping and 1000 random sequence additions, trees were inferred. Max trees were set to 5000, branches of zero length were collapsed, and all parsimonious trees were saved. Clade robustness was assessed using bootstrap (BT) analysis with 1000 replicates [60]. Descriptive tree statistics, tree length (TL), the consistency index (CI), the retention index (RI), the rescaled consistency index (RC), and the homoplasy index (HI) were calculated for each maximum parsimonious tree generated. The multiple sequence alignment was also analyzed using maximum likelihood (ML) in RAxML-HPC2 [61]. Branch support (BS) for ML analysis was determined by 1000 bootstrap replicates.

MrModeltest 2.3 [62] was used to determine the best-fit evolution model for each dataset for Bayesian inference (BI), which was performed using MrBayes 3.2.7a with a GTR+I+G model of DNA substitution and a gamma distribution rate variation across sites [63]. A total of four Markov chains were run for two runs from random starting trees for 1.2 million generations for ITS+nLSU (Figure 1) and 9 million generations for ITS (Figure 2) with trees and parameters sampled every 1000 generations. The first one-fourth of all of the generations were discarded as burn-ins. The majority-rule consensus tree of all of the remaining trees was calculated. Branches were considered significantly supported if they received a maximum likelihood bootstrap value (BS) of >70%, a maximum parsimony bootstrap value (BT) of >70%, or Bayesian posterior probabilities (BPP) of >0.95.

## 3. Results

### 3.1. Molecular Phylogeny

The ITS+nLSU (Figure 1) included sequences from 50 fungal samples representing 50 species. The dataset had an aligned length of 1919 characters, of which 1140 characters were constant, 206 were variable and parsimony uninformative, and 573 were parsimony informative. The maximum parsimony analysis yielded six equally parsimonious trees (TL = 3564, CI = 0.3533, HI = 0.6467, RI = 0.5235, and RC = 0.1849). The best model for the ITS+nLSU dataset estimated and applied in the Bayesian analysis was GTR+I+G. The Bayesian and ML analyses showed a similar topology to that of the MP analysis with split frequencies = 0.008056 (BI), and the effective sample size (ESS) average ESS (avg ESS) = 418.5. The phylogram based on the ITS+nLSU rDNA gene regions (Figure 1) includes four families within Hymenochaetales, which comprises six genera: *Fasciodontia*, *Hastodontia*, *Hyphodontia*, *Kneiffiella*, *Lyomyces*, and *Xylodon*, which indicated that three genera (*Fasciodontia*, *Lyomyces,* and *Xylodon*) fell into the family Schizoporaceae.

The ITS-only dataset (Figure 2) included sequences from 67 fungal specimens representing 61 taxa. The dataset had an aligned length of 582 characters, of which 233 characters were constant, 59 were variable and parsimony uninformative, and 290 were parsimony informative. Maximum parsimony analysis yielded 5000 equally parsimonious trees (TL = 2191, CI = 0.2743, HI = 0.7257, RI = 0.4560, and RC = 0.1251). The best model for the ITS dataset estimated and applied in the Bayesian analysis was GTR+I+G. The Bayesian and ML analyses resulted in a similar topology to that of the MP analysis with split frequencies = 0.022072 (BI), and the effective sample size (ESS) of the average ESS (avg ESS) = 1670.5. The phylogram inferred from the ITS sequences analysis (Figure 2) indicated that three new species grouped into genus *Xylodon*: the new species *X. grandineus* was a sister to *X. nesporii* (Bres.) Hjortstam and Ryvarden; *X. punctus* formed a monophyletic lineage and then grouped with *X. filicinus* Yurchenko and Riebesehl, *X**. hastifer* (Hjortstam and Ryvarden) Hjortstam and Ryvarden, *X. hyphodontinus* (Hjortstam and Ryvarden) Riebesehl, Yurchenko and G. Gruhn, and *X. tropicus*, while *X.*
*wenshanensis* was retrieved as a sister of *X. xinpingensis* C.L. Zhao and X. Ma.

### 3.2. Taxonomy

***Xylodon grandineus*** K.Y. Luo and C.L. Zhao, sp. nov. Figure 3 and Figure 4.

MycoBank no.: 843054.

**Holotype**—China, Yunnan Province, Yuxi, Xinping County, Mopanshan National Forestry Park. GPS coordinates: 24°53′ N, 101°57′ E; altitude: 2000 m asl. On fallen *Pinus armandii* branches, leg. C.L. Zhao, 19 January 2018, CLZhao 6425 (SWFC).

**Etymology**—***grandineus*** (Lat.): Referring to the hymenial surface grandinioid of the specimens.

**Basidiomata**—Annual, resupinate, adnate, soft coriaceous when fresh, coriaceous upon drying, up to 10 cm long, 1.5 cm wide, 100–200 µm thick. Hymenial surface grandinioid, without odor or taste when fresh, pale buff when fresh, pale buff to buff when dry. Sterile margin indistinct, cream to buff, 0.5–1 mm wide.

**Hyphal system**—Monomitic, generative hyphae with clamp connections, colorless, thick-walled, frequently branched, interwoven, 2–4 µm in diameter, IKI−, CB−; tissues unchanged in KOH; subhymenial hyphae densely covered by the crystals.

**Hymenium**—Cystidia subulate, colorless, thin-walled, smooth, 11–19 × 3–5 µm; basidia barreled, constricted, with four sterigmata and a basal clamp connection, 11–19 × 2.5–4 µm.

**Spores**—Basidiospores ellipsoid, colorless, thin-walled, smooth, with one oil drop inside, IKI−, CB−, 3–4.5(–5) × 2–3 µm, L = 3.69 µm, W = 2.46 µm, Q = 1.46–1.54 (*n* = 60/2).

**Additional specimen****examined (paratype)**—China, Yunnan Province, Wenshan, Pingba Town, Wenshan National Nature Reserve. GPS coordinates: 23°15′ N, 104°06′ E; altitude: 1600 m asl. On fallen angiosperm branches, leg. C.L. Zhao, 25 July 2019, CLZhao 16075 (SWFC).

***Xylodon punctus*** K.Y. Luo and C.L. Zhao, sp. nov. Figure 5 and Figure 6.

MycoBank no.: 843055.

**Holotype**—China, Yunnan Province, Honghe, Pingbian County, Daweishan National Nature Reserve. GPS coordinates: 23°42′ N, 103°30′ E; altitude: 1500 m asl. On fallen angiosperm branches, leg. C.L. Zhao, 1 August 2019, CLZhao 17691 (SWFC).

**Etymology**—***punctus*** (Lat.): Referring to the spotted hymenial surface of the specimens.

**Basidiomata**—Annual, resupinate, adnate, thin, membranous, very hard to separate from substrate, up to 12 cm long, 1.5 cm wide, 20–80 µm thick. Hymenial surface smooth, speckled distribution, white when fresh, white to slightly grey upon drying. Sterile margin indistinct, white, up to 1 mm wide.

**Hyphal system**—Monomitic, generative hyphae with clamp connections, colorless, thin- to thick-walled, occasionally branched, interwoven, 1–3 µm in diameter, IKI−, CB−; tissues unchanged in KOH.

**Hymenium**—Cystidia absent; basidia clavate, short and obtused, with four sterigmata and a basal clamp connection, 10–16 × 3.5–5.5 µm.

**Spores**—Basidiospores ellipsoid to broad ellipsoid, colorless, thin-walled, smooth, IKI−, CB−, (1.5–)2–4(–4.5) × 1.5–2.5(–3) µm, L = 2.71 µm, W = 1.98 µm, Q = 1.32–1.43 (*n* = 31/3).

**Additional specimen****s examined (paratypes)**—China, Yunnan Province, Honghe, Pingbian County, Daweishan National Nature Reserve. GPS coordinates: 23°42′ N, 103°30′ E; altitude: 1500 m asl. On fallen angiosperm branches, leg. C.L. Zhao, 1 August 2019, CLZhao 17908; CLZhao 17916 (SWFC).

***Xylodon wenshanensis*** K.Y. Luo and C.L. Zhao, sp. nov. Figure 7 and Figure 8.

MycoBank no.: 843056.

**Holotype**—China, Yunnan Province, Wenshan, Xichou County, Jiguanshan Forestry Park. GPS coordinates: 23°15′ N, 104°40′ E; altitude: 1500 m asl. On fallen angiosperm branches, leg. C.L. Zhao, 22 July 2019, CLZhao 15729 (SWFC).

**Etymology**—***wenshanensis*** (Lat.): Referring to the specimens’ provenance from the Wenshan locality.

**Basidiomata**—Annual, resupinate, thin, without odor and taste when fresh, coriaceous, up to 9 cm long, 2 cm wide, 50–100 µm thick. Hymenial surface grandinioid, cream when fresh, cream to slightly buff upon drying. Sterile margin indistinct, cream, about 1 mm wide.

**Hyphal system**—Monomitic, generative hyphae with clamp connections, colorless, thin- to thick-walled, frequently branched, interwoven, 2–3.5 µm in diameter; IKI−, CB−; tissues unchanged in KOH.

**Hymenium**—Cystidia of two types: (1) capitate cystidia in hymenium and subiculum, colorless, thin-walled, smooth, slightly constricted at the neck, with a globose head, 6–11 × 3–6.5 µm; (2) clavate cystidia, slightly sinuous, 10.5–20 × 2.5–5.5 µm; basidia clavate to subcylindrical, slightly sinuous, with four sterigmata and a basal clamped connection, 8–15.5 × 3–5 µm.

**Spores**—Basidiospores ellipsoid, colorless, thin-walled, smooth, IKI−, CB−, 3–5 × 2–3.5(–4) µm, L = 3.96 µm, W = 2.82 µm, Q = 1.35–1.47 (*n* = 120/4).

**Additional****specimens examined (paratypes)**—China, Yunnan Province, Wenshan, Xichou County, Xiaoqiaogou, Wenshan National Nature Reserve. GPS coordinates: 23°22′ N, 104°43′ E; altitude: 1500 m asl. On fallen angiosperm branches, leg. C.L. Zhao, 14 January 2019, CLZhao 10790 (SWFC); Jiguanshan Forestry Park. GPS coordinates: 23°15′ N, 104°40′ E; altitude: 1500 m asl. On fallen angiosperm branches, leg. C.L. Zhao, 22 July 2019, CLZhao 15718, CLZhao 15782 (SWFC).

## 4. Discussion

Many recently described wood-inhabiting fungi taxa have been reported in the subtropics and tropics, including those of the genus *Xylodon* [64,65,66,67,68,69,70], which were collected on rotten trunks and stumps of conifers and angiosperms, bamboo, and ferns [2,3,6,15,23,25,40,41,43,52,71,72,73,74,75,76,77,78,79,80,81,82,83]. The present study reports three new taxa of *Xylodon*: *X. grandineus*, *X. punctus,* and *X. wenshanensis*, based on a combination of morphological features and molecular evidence.

Phylogenetically, the molecular relationships of *Xylodon* and related genera located in *Hyphodontia* s.l. (Hymenochaetales), on the basis of the combined datasets of ITS, nLSU, and mt-SSU regions, indicated that seven families—Chaetoporellaceae Jülich, Coltriciaceae Jülich, Hymenochaetaceae Donk, Neoantrodiellaceae Y.C. Dai, B.K. Cui, Jia J. Chen and H.S. Yuan, Nigrofomitaceae Jülich, Oxyporaceae Zmitr. and Malysheva, and Schizoporaceae—were monophyletic lineages, which nested in the order Hymenochaetales, in which some genera grouped into *Hyphodontia* s.l. as independent genera, including *Xylodon* [41]. In the present study (Figure 1), four families in the order Hymenochaetales were analyzed by the ITS+nLSU data, which showed that the genus *Xylodon* nested into the family Schizoporaceae.

The ITS-based evolution phylogram for *Xylodon* and related species revealed four species—*X. cystidiatus* (A. David and Rajchenb.) Riebesehl and Langer; *X. hyphodontinus*; *X. serpentiformis* (Langer) Hjortstam and Ryvarden; and *X. subclavatus* (Yurchenko, H.X. Xiong and Sheng H. Wu) Riebesehl, Yurch. and Langer—were in the genus *Xylodon* [14]. In the current study (Figure 2), the three new species also nested into the genus *Xylodon*, in which *X. grandineus* was a sister to *X. nesporii*; X. *punctus* formed a monophyletic lineage and then grouped with *X. filicinus*, *X. hastifer*, *X. hyphodontinus,* and *X. tropicus*, while *X. wenshanensis* was retrieved as a sister to *X. xinpingensis*. However, morphologically, *Xylodon nesporii* can be delimited from *X. grandineus* has an odontioid hymenial surface and narrowly ellipsoid to cylindrical basidiospores (4.5–6 × 2–2.5 µm) [70]. *Xylodon filicinus* differs from *X. punctus* by its odontioid hymenial surface and larger, globose to subglobose basidiospores (4–5 × 4–4.5 µm) [14]. *Xylodon hastifer* could be delimited from *X. punctus* by its odontioid hymenial surface and larger, subglobose basidiospores (4.5–5 × 4–4.5 µm) [15]. *Xylodon hyphodontinus* is differs from *X. punctus* in its odontioid hymenial surface and globose to subglobose basidiospores (4.5–5 μm in diameter) [64]. *Xylodon tropicus* differs from *X. punctus* in its coriaceous basidiomata with a grandinioid hymenial surface and subglobose basidiospores [43]. *X**ylodon*
*xinpingensis* can be delimited from *X. wenshanensis* by its soft-membranaceous basidiomata, a reticulate hymenial surface, fusiform cystidia (19.5–31 × 2–6 µm), and larger subglobose basidiospores (5–6.4 × 3.5–5 µm) [52].

Morphologically, *Xylodon grandineus* is similar to *X*. *follis* Riebesehl, Yurchenko and Langer; *X*. *laceratus* C.L. Zhao; *X. macrosporus* C.L. Zhao and K.Y. Luo; *X. tropicus*; and *X. sinensis* C.L. Zhao and K.Y. Luo due to its the grandinioid hymenial surface. However, *Xylodon follis* differs from *X*. *grandineus* in its effused basidiomata with a cream-colored hymenial surface, capitate cystidia (17–30 × 4.5–9 µm), and larger globose to subglobose basidiospores (8–9.5 × 7–8.5 µm) [14]. *X**ylodon laceratus* differs from *X*. *grandineus* in its capitate cystidia (15.4–24.7 × 3.8–4.7 µm) and fusiform cystidia (20.3–26.8 × 5.3–6.4 µm) [43]. *X**ylodon macrosporus* differs from *X*. *grandineus* by having cystidia of three types: capitate cystidia (8–25.5 × 3–10 µm), cylindrical cystidia (44–79.5 × 3–6 µm), and cystidia (11–21 × 6–11 µm), as well as larger thick-walled basidiospores (8–10.5 × 7.5–9 µm) [40]. *Xylodon tropicus* can be delimited from *X*. *grandineus* by its buff to pale brown hymenial surface; absent cystidia; and subglobose, slightly thick-walled basidiospores [43]. *X**ylodon sinensis* is distinguishable from *X*. *grandineus* by its buff to brown hymenial surface, fusiform cystidia, and subglobose basidiospores [40].

*Xylodon grandineus* resembles *X*. *attenuatus* Spirin and Viner; *X**. borealis* (Kotir. and Saaren.) Hjortstam and Ryvarden; *X*. *bresinskyi* (Langer) Hjortstam and Ryvarden; *X**. dimiticus* (Jia J. Chen and L.W. Zhou) Riebesehl and E. Langer; and *X. vesiculosus* Yurchenko, Nakasone and Riebesehl with its ellipsoid basidiospores. However, *Xylodon attenuatus* differs from *X*. *grandineus* in its cream-colored, grandinioid to odontoid hymenial surface with rather regularly arranged projections; cystidia of two types: subcapitate or capitate cystidia (13.5–25.1 × 3.5–5 µm) and hyphoid cystidia (16–38.3 × 2.8–4.5 µm); and wider basidiospores (4.1–5.5 × 3.4–4.5 µm) [36]. *X**ylodon borealis* differs from *X*. *grandineus* by having effused basidiomata; cystidia of two types: capitate cystidia (20–50 × 4–6 µm) and slender hypha-like cystidia (40–70 × 3–5 µm); and larger basidiospores (4.5–5.5 × 3.5–4 µm) [4]. *X**ylodon bresinskyi* differs from *X*. *grandineus* in its poroid hymenial surface with rudimentary console shaping and larger basidiospores (4.5–5.5 × 3–3.5 µm) [84]. *X**ylodon dimiticus* is distinguishable from *X*. *grandineus* by its poroid hymenial surface with angular pores (2–4 per mm) and absent cystidia [28]. *Xylodon*
*vesiculosus* can be delimited from *X*. *grandineus* by its membranaceous basidiomata with an odontioid hymenial surface and larger basidiospores (5.3–6.3 × 3–4 µm) [14].

*Xylodon punctus* is similar to *X. acystidiatus* Xue W. Wang and L.W. Zhou, *X. gossypinus* C.L. Zhao and K.Y. Luo, *X. montanus* C.L. Zhao, and *X. nudisetus* (Warcup and P.H.B. Talbot) Hjortstam and Ryvarden in having a smooth hymenial surface. However, *Xylodon acystidiatus* differs from *X. punctus* by having brittle basidiomata with a cracked hymenial surface and larger basidiospores (4.7–5.3 × 2.7–3.7 µm) [41]. *Xylodon gossypinus* differs from *X. punctus* in its cotton hymenial surface and wider basidiospores (3–5.5 × 2.5–4 μm) [40]. *Xylodon montanus* can be delimited from *X. punctus* by its absence of a speckled distribution on the hymenial surface and wider basidiospores (3.9–5.3 × 3.2–4.3 μm) [43]. *Xylodon nudisetus* differs from *X. punctus* in having larger basidiospores (4.5–6 × 3–4.5 µm) [4].

*Xylodon punctus* resembles *X. bambusinus* C.L. Zhao and X. Ma; *X**. mussooriensis* Samita, Sanyal and Dhingra ex L.W. Zhou and T.W. May; *X**. rhododendricola* Xue W. Wang and L.W. Zhou; *X. pruinosus* (Bres.) Spirin and Viner; and *X. ussuriensis* Viner in having ellipsoid to broad ellipsoid basidiospores. However, *Xylodon*
*bambusinus* is distinguished from *X. punctus* by its ceraceous basidiomata with a grandinoid hymenial surface and larger basidiospores (4–5 × 2.6–3.7 µm) [52]. *X**ylodon mussooriensis* differs from *X. punctus* by the presence of an odontioid hymenial surface and larger basidiospores (5.2–5.8 × 3.1–3.5 µm) [41]. *X**ylodon rhododendricola* differs from *X. punctus* in having an odontioid hymenial surface and larger basidiospores (4.8–6.5 × 3.8–5.1 µm) [41]. *X**ylodon pruinosus* differs from *X. punctus* in having a grandinioid to odontoid hymenial surface with greyish-white or pale cream-colored and larger, clearly thick-walled basidiospores (4.5–5.9 × 3.7–4.8 µm) [36]. *X**ylodon ussuriensis* is distinguished from *X. punctus* by its grandinioid to odontoid hymenial surface with larger, pale ochraceous, and clearly thick-walled basidiospores (5.1–6 × 3.8–4.6 µm) [36].

*Xylodon wenshanensis* is similar to *X. laceratus, X. macrosporus*, *X. sinensis*, *X. victoriensis* Xue W. Wang and L.W. Zhou, and *X*. *yarraensis* Xue W. Wang and L.W. Zhou in having a grandinioid hymenial surface. However, *Xylodon laceratus* is distinguished from *X. wenshanensis* by its capitate cystidia (15.4–24.7 × 3.8–4.7 µm) and fusiform cystidia (20.3–26.8 × 5.3–6.4 µm) [43]. *Xylodon macrosporus* is differentiated from *X. wenshanensis* in having three types cystidia and larger, thick-walled basidiospores (8–10.5 × 7.5–9 µm) [40]. *Xylodon sinensis* differs from *X. wenshanensis* in its fusiform cystidia (10–21 × 3–6 µm) and subglobose basidiospores (3–5 × 2.5–4 µm) [40]. *Xylodon victoriensis* can be delimited from *X. wenshanensis* by its brittle basidiomata with a cracked hymenophore, leptocystidia (30–40 × 4.5–5 μm), and globose to subglobose basidiospores (3.8–4.6 × 3.2–3.7 μm) [41]. *Xylodon yarraensis* is different from *X. wenshanensis* in its cracked and brittle basidiomata and capitate cystidia (25–30 × 2.5–3.5 µm) [41].

*Xylodon wenshanensis* resembles *X. asper* (Fr.) Hjortstam and Ryvarden; *X. flaviporus* (Berk. and M.A. Curtis ex Cooke) Riebesehl and Langer; *X. ovisporus* (Corner) Riebesehl and Langer; *X. pseudolanatus* Nakasone, Yurchenko and Riebesehl; and *X. rimosissimus* (Peck) Hjortstam and Ryvarden in having capitate cystidia. However, *Xylodon asper* is different from *X. wenshanensis* in having an odontioid hymenial surface with scattered aculei and larger basidiospores (5–6 × 3.5–4 μm) [4]. *Xylodon flaviporus* is distinguished from *X. wenshanensis* by its poroid hymenial surface with deep pores (up to 2 mm) and a pseudodimitic hyphal system [15]. *Xylodon ovisporus* is differentiated from *X. wenshanensis* by having a poroid hymenophore with pinkish-cream or buff hymenial surface [19]. *Xylodon pseudolanatus* differs from *X. wenshanensis* by its emebranaceous basidiomata, odontioid hymenial surface, and longer basidiospores (5–6 × 3–3.5 µm) [14]. *Xylodon rimosissimus* can be delimited from *X. wenshanensis* by its subceraceous basidiomata with a dense odontioid hymenial surface and larger basidiospores (5–6 × 3.5–4 μm) [4].

The macromorphology of the basidiomata and hymenophore construction do not reflect monophyletic groups based on a higher-level phylogenetic classification of polypores [82]. The current phylogeny (Figure 2) shows that the morphological characteristics do not follow the phylogenetic grouping of different taxa in *Xylodon* based on the ITS datasets.

Wood-inhabiting fungi are a characteristic group of Basidiomycota, which has a number of corticioid, poroid, and hydnoid genera based on the results of morphological, phylogenetic, and cytological studies in China [8,9]. To date, thirty-six species of *Xylodon* have been recorded in China [12,14,33,36,40,41,43,46,52,83], but the species diversity of *Xylodon* is still not well known in China, especially in the country’s subtropical and tropical areas. This paper enriches our knowledge of fungal diversity in this area, and it is likely that more new taxa will be found with further fieldwork and molecular analyses.

## Figures and Tables

**Figure 1 jof-08-00405-f001:**
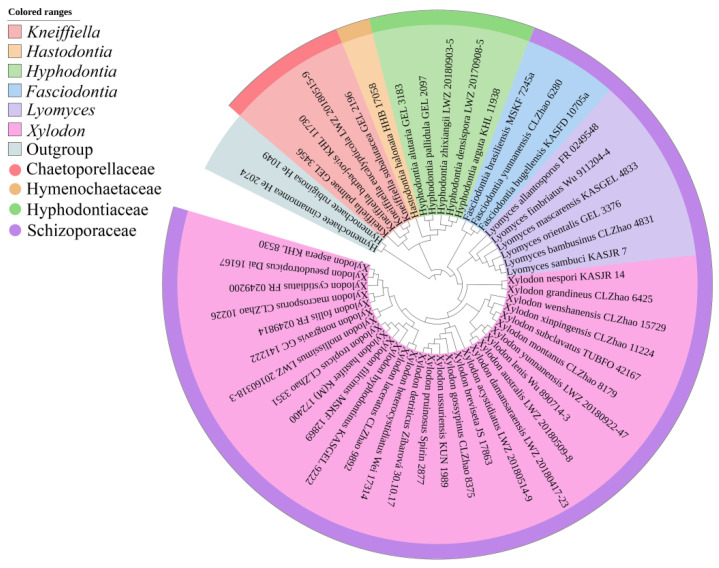
Maximum parsimony strict consensus tree illustrating the phylogeny of *Xylodon* and related genera in the order Hymenochaetales based on ITS+nLSU sequences. The families and genera represented by each color are indicated in the upper left of the phylogenetic tree.

**Figure 2 jof-08-00405-f002:**
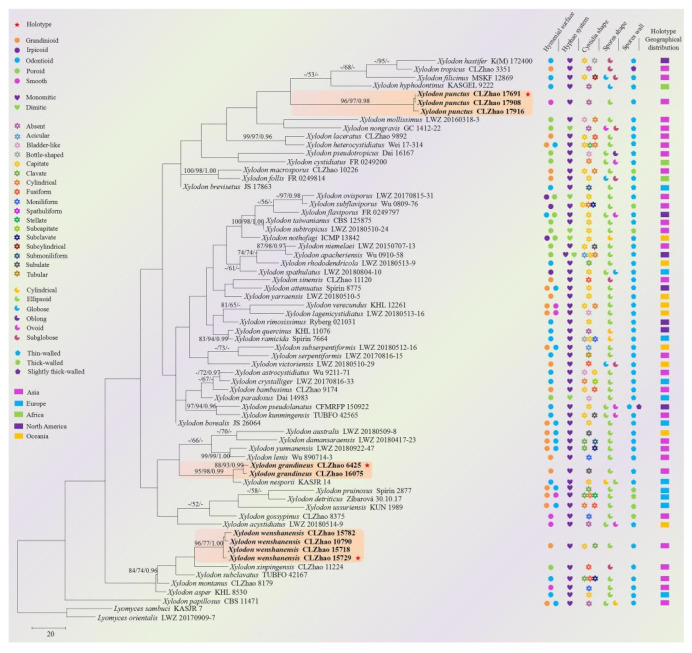
Maximum parsimony strict consensus tree illustrating the phylogeny of three new species in *Xylodon* based on ITS sequences. Branches are labeled with a maximum likelihood bootstrap value > 70%, a parsimony bootstrap value > 50%, and Bayesian posterior probabilities > 0.95, respectively. The new species are in bold. The red stars representative holotypes.

**Figure 3 jof-08-00405-f003:**
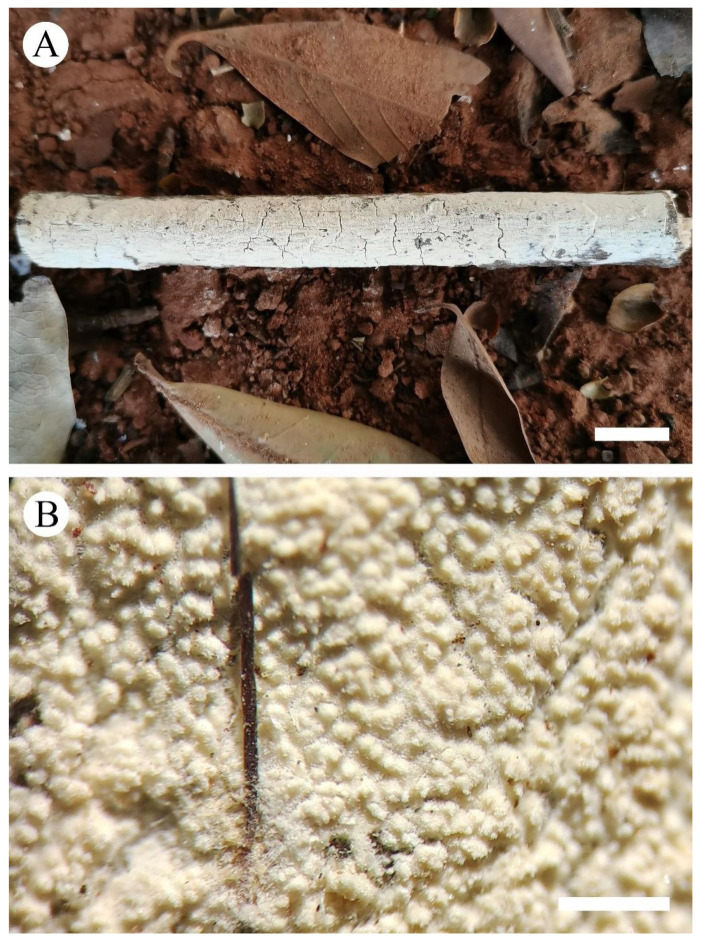
Basidiomata of *Xylodon grandineus* (holotype): the front of the basidiomata (**A**), characteristic hymenophore (**B**). Bars: (**A**) = 1 cm and (**B**) = 1 mm.

**Figure 4 jof-08-00405-f004:**
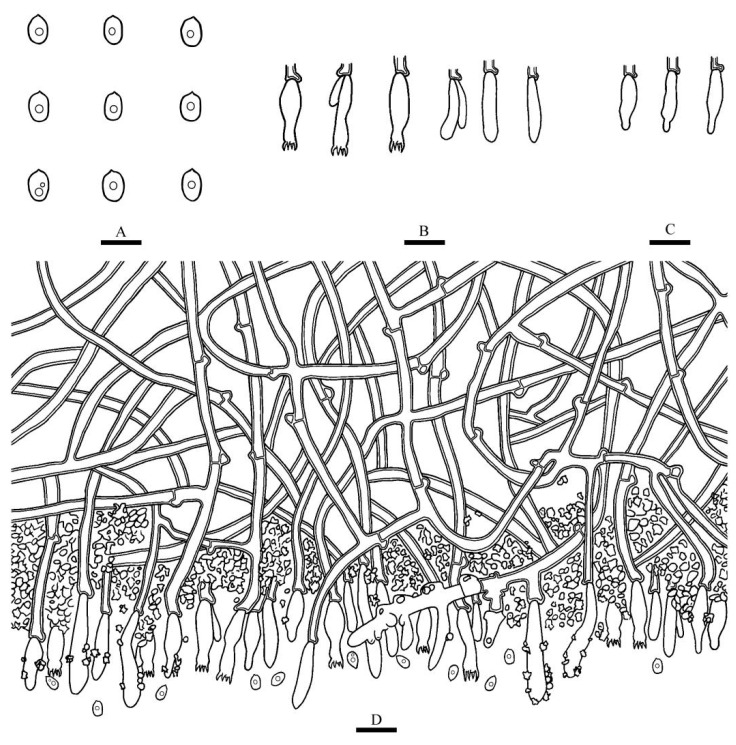
Microscopic structures of *Xylodon grandineus* (holotype): basidiospores (**A**), basidia and basidioles (**B**), subulate cystidia (**C**), a section of the hymenium (**D**). Bars: (**A**) = 5 μm, (**B**–**D**) = 10 µm.

**Figure 5 jof-08-00405-f005:**
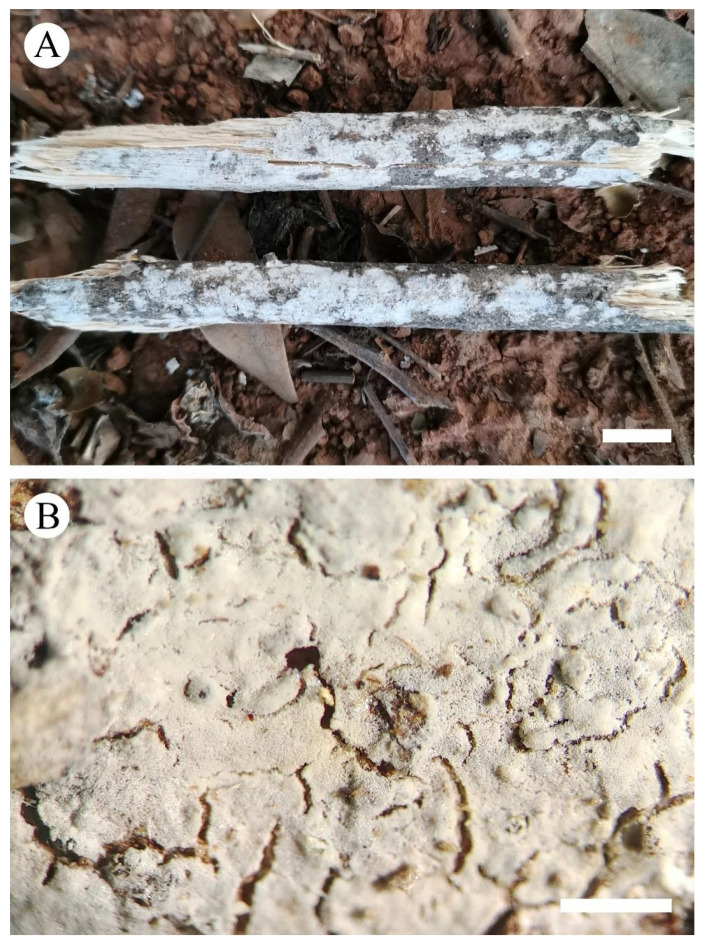
Basidiomata of *Xylodon punctus* (holotype): the front of the basidiomata (**A**), characteristic hymenophore (**B**). Bars: (**A**) = 1 cm and (**B**) = 1 mm.

**Figure 6 jof-08-00405-f006:**
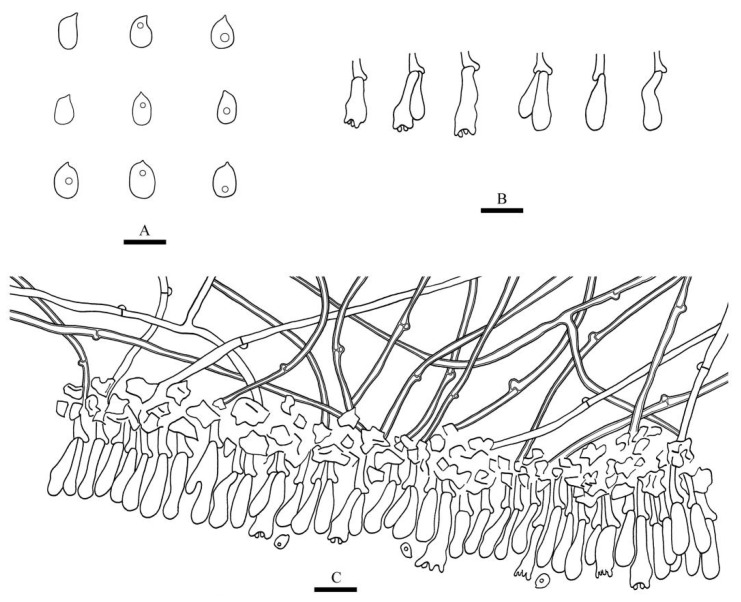
Microscopic structures of *Xylodon punctus* (holotype): basidiospores (**A**), basidia and basidioles (**B**), a section of the hymenium (**C**). Bars: (**A**) = 5 μm, (**B**,**C**) = 10 µm.

**Figure 7 jof-08-00405-f007:**
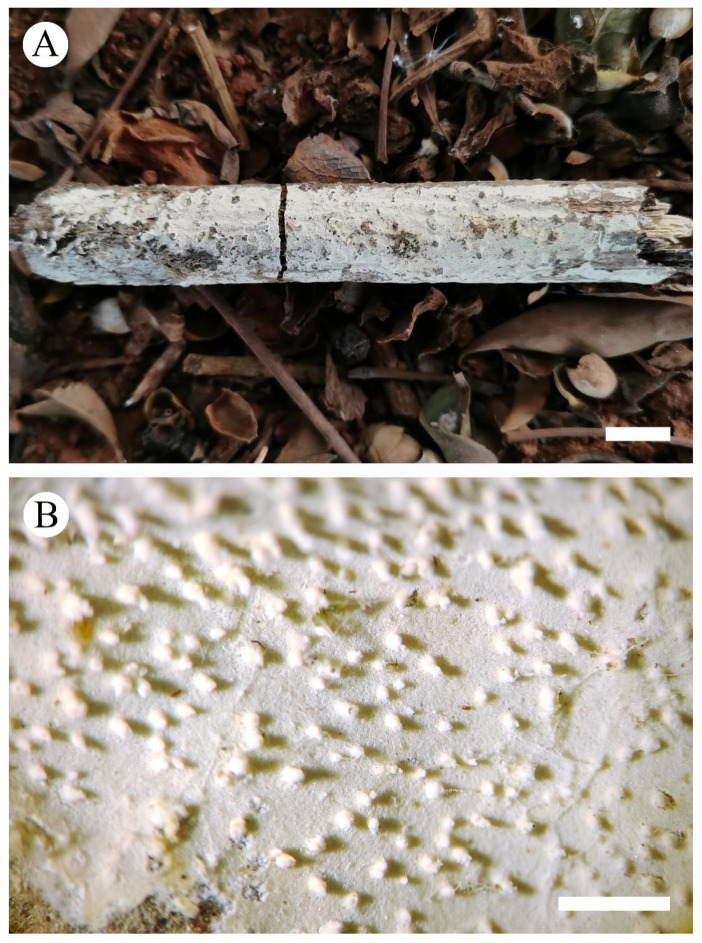
Basidiomata of *Xylodon wenshanensis* (holotype): the front of the basidiomata (**A**), characteristic hymenophore (**B**). Bars: (**A**) = 1 cm and (**B**) = 1 mm.

**Figure 8 jof-08-00405-f008:**
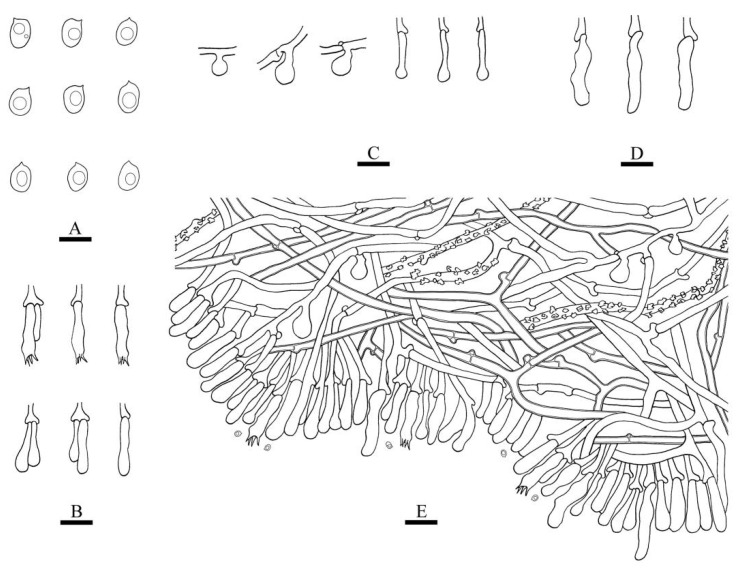
Microscopic structures of *Xylodon wenshanensis* (holotype): basidiospores (**A**), basidia and basidioles (**B**), capitate cystidia in subiculum and hymenium (**C**), clavate cystidia (**D**), a section of the hymenium (**E**). Bars: (**A**) = 5 μm, (**B**–**E**) = 10 µm.

**Table 1 jof-08-00405-t001:** List of species, specimens, and GenBank accession numbers of sequences used in this study.

Species Name	Specimen No.	GenBank Accession No.		References	Country
		ITS	nLSU		
*Fasciodontia brasiliensis*	MSKF 7245a	MK575201	MK598734	[46]	Brazil
*F. bugellensis*	KASFD 10705a	MK575203	MK598735	[46]	France
*F. yunnanensis*	CLZhao 6280	MK811275	MZ146327	[47]	China
*Hastodontia halonata*	HHB 17058	MK575207	MK598738	[46]	Mexico
*Hymenochaete cinnamomea*	He 2074	KU975460	KU975500	Unpublished	China
*Hym. rubiginosa*	He 1049	JQ716407	JQ279667	[48]	China
*Hyphodontia alutaria*	GEL 3183	DQ340318	DQ340373	Unpublished	Germany
*Hyp. arguta*	KHL 11938	EU118632	EU118633	[49]	Sweden
*Hyp. densispora*	LWZ 20170908-5	MT319426	MT319160	[41]	China
*Hyp. pallidula*	KASGEL 2097	DQ340317	DQ340372	Unpublished	Germany
*Hyp. zhixiangii*	LWZ 20180903-5	MT319423	MT319158	[41]	China
*Kneiffiella barba-jovis*	KHL 11730	DQ873609	DQ873610	[14]	Sweden
*K. eucalypticola*	LWZ 20180515-9	MT319411	MT319143	[41]	Australia
*K. palmae*	KASGEL 3456	DQ340333	DQ340369	[46]	China
*K. subalutacea*	GEL 2196	DQ340341	DQ340362	[46]	Norway
*Lyomyces allantosporus*	FR 0249548	KY800397	KY795963	[13]	Réunion
*L. bambusinus*	CLZhao 4831	MN945968	MW264919	[50]	China
*L. fimbriatus*	Wu 911204-4	MK575210	MK598740	[46]	China
*L. mascarensis*	KASGEL 4833	KY800399	KY795964	[46]	Réunion
*L. orientalis*	LWZ 20170909-7	MT319436	MT319170	[41]	China
*L. sambuci*	KASJR 7	KY800402	KY795966	[13]	Germany
*Xylodon acystidiatus*	LWZ 20180514-9	MT319474	MT319211	[41]	Australia
*X. apacheriensis*	Wu 0910-58	KX857797		[11]	China
*X. asper*	KHL 8530	AY463427	AY586675	[51]	Sweden
*X. astrocystidiatus*	Wu 9211-71	JN129972		[26]	China
*X. attenuatus*	Spirin 8775	MH324476		[36]	America
*X. australis*	LWZ 20180509-8	MT319503	MT319248	[41]	China
*X. bambusinus*	CLZhao 9174	MW394657		[52]	China
*X. borealis*	JS 26064	AY463429		[51]	Norway
*X. brevisetus*	JS 17863	AY463428	AY586676	[51]	Norway
*X. crystalliger*	LWZ 20170816-33	MT319521		[41]	China
*X. cystidiatus*	FR 0249200	MH880195	MH884896	[14]	Réunion
*X. damansaraensis*	LWZ 20180417-23	MT319499	MT319244	[41]	Malaysia
*X. detriticus*	Zíbarová 30.10.17	MH320793	MH651372	[36]	Czech Republic
*X. filicinus*	MSKF 12869	MH880199	NG067836	[14]	China
*X. flaviporus*	FR 0249797	MH880201		[14]	Réunion
*X. follis*	FR 0249814	MH880204	MH884902	[14]	Réunion
*X. grandineus*	CLZhao 6425 *	OM338090	OM338099	Present study	China
*X. grandineus*	CLZhao 16075	OM338091	OM338100	Present study	China
*X. gossypinus*	CLZhao 8375	MZ663804	MZ663813	[40]	China
*X. hastifer*	K(M) 172400	NR166558		[12]	USA
*X. heterocystidiatus*	Wei 17-314	MT731753	MT731754	Unpublished	China
*X. hyphodontinus*	KASGEL 9222	MH880205	MH884903	[14]	Kenya
*X. kunmingensis*	TUBFO 42565	MH880198		[14]	China
*X.laceratus*	CLZhao 9892	OL619258		[43]	China
*X. lagenicystidiatus*	LWZ 20180513-16	MT319634		[41]	Australia
*X. lenis*	Wu 890714-3	KY081802		[12]	China
*X. macrosporus*	CLZhao 10226	MZ663809	MZ663817	[40]	China
*X. mollissimus*	LWZ 20160318-3	KY007517	MT319347	[41]	China
*X.montanus*	CLZhao 8179	OL619260		[43]	China
*X. nesporii*	LWZ 20180921-35	MT319655	MT319238	[41]	China
*X. niemelaei*	LWZ 20150707-13	MT319630		[41]	China
*X. nongravis*	GC 1412-22	KX857801	KX857818	[11]	China
*X. nothofagi*	ICMP 13842	AF145583		[53]	China
*X. ovisporus*	LWZ 20170815-31	MT319666		[41]	China
*X. papillosus*	CBS 114.71	MH860026		[54]	The Netherlands
*X. paradoxus*	Dai 14983	MT319519		[41]	China
*X. pruinosus*	Spirin 2877	MH332700		[36]	Estonia
*X. pseudolanatus*	FP 150922	MH880220		[14]	Belize
*X. pseudotropicus*	Dai 16167	MT319509	MT319255	[41]	China
*X. punctus*	CLZhao 17691 *	OM338092	OM338101	Present study	China
*X. punctus*	CLZhao 17908	OM338093		Present study	China
*X. punctus*	CLZhao 17916	OM338094	OM338102	Present study	China
*X. quercinus*	KHL 11076	KT361633		[51]	Sweden
*X. ramicida*	Spirin 7664	NR138013		Unpublished	USA
*X. rhododendricola*	LWZ 20180513-9	MT319621		[41]	Australia
*X. rimosissimus*	Ryberg 021031	DQ873627		[55]	Sweden
*X. serpentiformis*	LWZ 20170816-15	MT319673		[41]	China
*X. sinensis*	CLZhao 11120	MZ663811		[40]	China
*X. spathulatus*	LWZ 20180804-10	MT319646		[41]	China
*X. subclavatus*	TUBFO 42167	MH880232		[14]	China
*X. subflaviporus*	Wu 0809-76	KX857803		[11]	China
*X. subserpentiformis*	LWZ 20180512-16	MT319486		[41]	Australia
*X. subtropicus*	LWZ 20180510-24	MT319541		[41]	China
*X. taiwanianus*	CBS 125875	MH864080		[54]	The Netherlands
*X. tropicus*	CLZhao 3351	OL619261	OL619269	[43]	China
*X. ussuriensis*	KUN 1989	NR166241		Unpublished	USA
*X. verecundus*	KHL 12261	DQ873642		[55]	Sweden
*X. victoriensis*	LWZ 20180510-29	MT319487		[41]	Australia
*X. wenshanensis*	CLZhao 10790	OM338095	OM338103	Present study	China
*X. wenshanensis*	CLZhao 15718	OM338096		Present study	China
*X. wenshanensis*	CLZhao 15729 *	OM338097	OM338104	Present study	China
*X. wenshanensis*	CLZhao 15782	OM338098	OM338105	Present study	China
*X. xinpingensis*	CLZhao 11224	MW394662	MW394654	[52]	China
*X. yarraensis*	LWZ 20180510-5	MT319639		[41]	Australia
*X. yunnanensis*	LWZ 20180922-47	MT319660	MT319253	[41]	China

* is shown holotype.

## Data Availability

Publicly available datasets were analyzed in this study. This data can be found here: https://www.ncbi.nlm.nih.gov/; https://www.mycobank.org/page/Simple%20names%20search; http://purl.org/phylo/treebase, submission ID 29411; accessed on 17 February 2022.

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
