# Peer review of "Phylogenetic and Taxonomic Analyses of Three New Wood-Inhabiting Fungi of Xylodon (Basidiomycota) in a Forest Ecological System"

_jof, 2022, doi:10.3390/jof8040405_

Round 1

Reviewer 1 Report

The paper is okay.

I only found 3 grammatical errors:

line 41: change `it causes´ to `they cause´,

line 79: remove 1 `the´,

line 83: change `is´ to `are´.

The words in Fig. 1 are hardly decipherable.

Olaf Schmidt

Author Response

Dear Reviewer, please see the attachment, thank you.

Reviewer 2 Report

The article introduces Three New Wood-inhabiting Fungi of Xylodon (Basidiomycota) in Forest Ecological System which is quite interesting. It can be polished by incorporation of a dichotomous key consisting of closely related taxa. Similarly, concluding remarks can be improved.

There are few English language errors which needs to be corrected (indicated below)

line

16: morphogy….misspelled…morphology

29: white-rotting fungi? Is it white rot fungi or wood rotting fungi

42:  To carry out the genome evolution and reconstruction of phylogenetic relationships… rephrase please

43: fungi have served as models for using genome scale data to embrace?   of the molecular systematics by mycologists

45: in the more basal branches of the topology tree….in the more basal branches of the tree topology

52: and presence of types cystidia…presence of different types of cystidia

61-63 was employed by the mycologists on account of lacking of sequences for many taxa,  which leaded to the molecular data were not to support the separation these genera  clearly…rephrase please

63: tow clades….two

86 revealed that X. tropicus was retrieved as a sister to X. hastifer…sister species

100 Macromorphological descriptions and micromorphological data were followed  previous study…rephrase please

108 The PCR procedure for ITS and nLSU was following from previous study…rephrase please

124    followed previous study…delete these words

Author Response

(The authors gave the same response as above.)

Reviewer 3 Report

This manuscript has too bad grammar and I can doubt the meaning and gramatic correctness of each sentence. I labelled only grammar remarks in Introduction, but  I noticed it all over the manuscript. Moreover, I doubt if phylogenetic part is resonable only with ribosomal DNA, there is too much attention to this aspect but the data are shallow for it. The valuable part is description of three species, so it might be resubmitted if family and above genus phylogeny is not aim but only is discussed. Aims has to be also better defined. 

Author Response

(The authors gave the same response as above.)

Round 2

Reviewer 3 Report

"Polyphasic Identification Employing Phylogenetic and Taxonomic Analyses " is just a sofisticated way to say this is standard taxonomic paper based on two ribosomal markers and resulting in three new species. This is the only significance of the paper and if authors do not qustion the higher phylogeny in the abstract I would not complain. My major concern one was bad English, text structure and style. I stopped reading the manuscript after decisive explanatrion how bad is it in the introduction and authors made almost no change in other parts of the manuscript. My second major concern is how much authors read from the ribosomal DNA (actually this is a longer but only one genetic marker). The circular tree is interpreted as prove that Xylodon belongs to Schizoporaceae but there is no Schizopora (the type of family) included and neither it is discussed. Because authors did weak effort to improve I am recommending again to reject it. This study has no more than three new species based on ribosomal DNA and bad English, not worse for high IF journal. 

Author Response

Dear Reviewer, 

We are very grateful to you for your patient comments on our manuscript.
